# The effects of different footprint sizes and cloud algorithms on the top-of-atmosphere radiative flux calculation from the Clouds and Earth's Radiant Energy System (CERES) instrument on Suomi National Polar-orbiting Partnership (NPP)

WENYING SU, [*]

*Science Directorate, NASA Langley Research Center, Hampton, Virginia*

LUSHENG LIANG, WALTER F. MILLER, VICTOR E. SOTHCOTT

*Science Systems & Applications, Inc., Hampton, Virginia*

[*]*Corresponding author address:* Wenying Su, MS420, NASA Langley Research Center, Hampton, VA 23681.

E-mail: Wenying.Su-1@nasa.gov

# ABSTRACT

Only one CERES instrument is onboard the Suomi NPP and it has been placed in cross-track mode since launch, it is thus not possible to construct a set of angular distribution models (ADMs) specific for CERES on NPP. Edition 4 Aqua ADMs are used for flux inversions for CERES-NPP measurements. However, the footprint size of CERES-NPP is greater than that of CERES-Aqua, as the altitude of the NPP orbit is higher than that of the Aqua orbit. Furthermore, cloud retrievals from the Visible Infrared Imaging Radiometer Suite (VIIRS) and the Moderate Resolution Imaging Spectroradiometer (MODIS), the imagers sharing the spacecrafts with CERES-NPP and CERES-Aqua, are also different. To quantify the flux uncertainties due to the footprint size difference between CERES-Aqua and CERES-NPP, and due to both the footprint size difference and cloud property difference, a simulation is designed using the MODIS pixel level data which are convolved with the CERES-Aqua and CERES-NPP point spread functions into their respective footprints. The simulation is designed to isolate the effects of footprint size and cloud property differences on flux uncertainty from calibration and orbital differences between CERES-NPP and CERES-Aqua. The footprint size difference between CERES-Aqua and CERES-NPP introduces instantaneous flux uncertainties in monthly gridded CERES-NPP of less than 4.0 $\mathrm{Wm^{-2}}$ for SW, and less than 1.0 $\mathrm{Wm^{-2}}$ for both daytime and nighttime LW. The global monthly mean instantaneous SW flux from simulated CERES-NPP has a low bias of 0.4 $\mathrm{Wm^{-2}}$ when compared to simulated CERES-Aqua, and the root-mean-square (RMS) error is 2.2 $\mathrm{Wm^{-2}}$ between them; the biases of daytime and nighttime LW flux are close to zero with RMS errors of 0.8 $\mathrm{Wm^{-2}}$ and 0.2 $\mathrm{Wm^{-2}}$. These uncertainties are within the uncertainties of CERES ADMs. When both footprint size and cloud property (cloud fraction and optical depth) differences are considered, the uncertainties of monthly gridded CERES-NPP SW flux can be up to 20 $\mathrm{Wm^{-2}}$ in the Arctic regions where cloud optical depth retrievals from VIIRS differ significantly from MODIS. The global monthly mean instantaneous SW flux from simulated CERES-NPP has a high bias of 1.1 $\mathrm{Wm^{-2}}$ and the RMS error increases to 5.2 $\mathrm{Wm^{-2}}$. LW

flux shows less sensitivity to cloud property differences than SW flux, with the uncertainties of about 2 Wm$^{-2}$ in monthly gridded LW flux, and the RMS errors of global monthly mean daytime and nighttime fluxes increase only slightly. These results highlight the importance of consistent cloud retrieval algorithms to maintain the accuracy and stability of the CERES climate data record.

# 1.  Introduction

The Clouds and Earth's Radiant Energy System (CERES) project has been providing data products crucial to advancing our understanding of the effects of clouds and aerosols on radiative energy within the Earth-atmosphere system. CERES data are used by the science community to study the Earth's energy balance (e.g., Trenberth et al. 2009; Kato et al. 2011; Loeb et al. 2012; Stephens et al. 2012), aerosol direct radiative effects (e.g., Satheesh and Ramanathan 2000; Zhang et al. 2005; Loeb and Manalo-Smith 2005; Su et al. 2013), aerosol-cloud interactions (e.g., Loeb and Schuster 2008; Quaas et al. 2008; Su et al. 2010b), and to evaluate global general circulation models (e.g., Pincus et al. 2008; Su et al. 2010a; Wang and Su 2013; Wild et al. 2013).

Six CERES instruments have flown on four different satellites thus far. CERES pre-Flight Model (FM) on Tropical Rainfall Measuring Mission (TRMM) was launched on November 27, 1997 into a 350-km circular precessing orbit with a 35° inclination angle and flew together with the Visible and Infrared Scanner (VIRS). CERES instruments (FM1 and FM2) on Terra were launched on December 18, 1999 into a 705-km sun-synchronous orbit with a 10:30 a.m. equatorial crossing time. CERES instruments (FM3 and FM4) on Aqua satellite were launched on May 4, 2002 into a 705-km sun-synchronous orbit with a 1:30 p.m. equatorial crossing time. CERES on Terra and Aqua flies alongside Moderate-Resolution Imaging Spectroradiometer (MODIS). CERES instrument (FM5) was launched onboard Suomi NPP (hereafter referred to as NPP) on October 28, 2011 into a 824-km sun-synchronous orbit with a 1:30 p.m. equatorial crossing time and flies alongside the Visible Infrared Imaging Radiometer Suite (VIIRS). As the orbit altitudes differ among these satellites, the spatial resolutions of CERES instruments also vary from each other. TRMM has the lowest orbit altitude and offers the highest spatial resolution of CERES measurements, about 10 km at nadir; the spatial resolution of CERES on Terra and Aqua is about 20 km at nadir; and is about 24 km at nadir for NPP as it has the highest orbit altitude.

The CERES instrument consists of a three-channel broadband scanning radiometer (Wielicki

et al. 1996). The scanning radiometer measures radiances in shortwave (SW, 0.3-5 $\mu m$), window (WN, 8-12 $\mu m$), and total (0.3-200 $\mu m$) channels. The longwave (LW) component is derived as the difference between total and SW channels. These measured radiances ($I$) at a given sun-Earth-satellite geometry are converted to outgoing reflected solar and emitted thermal TOA radiative fluxes ($F$) as:

$$F(\theta_0) = \frac{\pi I(\theta_0, \theta, \phi)}{R_j(\theta_0, \theta, \phi)}. \tag{1}$$

where $\theta_0$ is the solar zenith angle, $\theta$ is the CERES viewing zenith angle, $\phi$ is the relative azimuth angle between CERES and the solar plane, and $R_j(\theta_0, \theta, \phi)$ is the anisotropic factors for scene type $j$. Here scene type is a combination of variables (e.g., surface type, cloud fraction, cloud optical depth, cloud phase, aerosol optical depth, precipitable water, lapse rate, etc) that are used to group the data to develop distinct angular distribution models (ADMs). Note the SW ADMs are developed as a function of $\theta_0, \theta, \phi$ for each scene type, whereas the LW ADMs are a weak function of $\theta_0$ and $\phi$ and are developed only as a function of $\theta$ (Loeb et al. 2005; Su et al. 2015a).

To facilitate the construction of ADMs, there are pairs of identical CERES instruments on both Terra and Aqua. At the beginning of these missions one of the instruments on each satellite was always placed in a rotating azimuth plane (RAP) scan mode, while the other one was placed in cross-track mode to provide spatial coverage. When in RAP mode, the instrument scans in elevation as it rotates in azimuth, thus acquiring radiance measurements from a wide range of viewing combinations. There are about 60 months of RAP data collected on Terra and about 32 months of RAP data collected on Aqua. CERES instruments fly alongside high-resolution imagers, which provide accurate scene type information within the CERES footprints. Cloud and aerosol retrievals based upon high-resolution imager measurements are averaged over the CERES footprints by accounting for the CERES point spread function (PSF, Smith 1994) and are used for scene type classification. Similarly, spectral radiances from MODIS/VIIRS observations are averaged over the CERES footprints weighted by the CERES PSF. Surface types are obtained from the International Geosphere

Biosphere Program (IGBP, Loveland and Belward 1997) global land cover data set. Fresh snow and sea ice surface types are derived from a combination of the National Snow and Ice Data Center (NSIDC) microwave snow/ice map and the National Environmental Satellite, Data and Information Service (NESDIS) snow/ice map. NESDIS uses imager data to identify snow and sea ice and provide snow and sea ice information near the coast, whereas NSIDC does not provide microwave retrievals within 50 km of the coast.

TRMM ADMs were developed using 9 months of CERES observations and the scene identification information retrieved from VIRS observations (Loeb et al. 2003). Terra ADMs and Aqua ADMs were developed separately using multi-year CERES Terra and Aqua measurements in RAP mode and in cross-track mode using the scene identification information from Terra MODIS and Aqua MODIS (Loeb et al. 2005; Su et al. 2015a). The high-resolution MODIS imager provides cloud conditions for every CERES footprint. The cloud algorithms developed by the CERES cloud working group retrieve cloud fraction, cloud optical depth, cloud phase, cloud top and effective temperature/pressure (among other variables) based on MODIS pixel-level measurements (Minnis et al. 2010). These pixel-level cloud properties are spatially and temporally matched with the CERES footprints and are used to select the scene-dependent ADMs to convert the CERES measured radiances to fluxes (Eq.1). The spatial matching criterion used is 1 km. The temporal matching criterion used is less than 20 seconds when CERES is in cross-track mode, and less than 6 minutes when CERES is in RAP mode.

There is only one CERES instrument on NPP and it has been placed in cross-track scan mode since launch, it is thus not feasible to develop a specific set of ADMs for CERES on NPP. Currently, the Edition 4 Aqua ADMs (Su et al. 2015a) are used to invert fluxes for the CERES measurements on NPP. The CERES footprint size on NPP is larger than that on Aqua. As pointed out by Di Girolamo et al. (1998), the nonreciprocal behavior of the radiation field depends on measurement resolution, which means the ADMs do too. They concluded that ADMs should be applied only to data of the same resolution as the

data used to derive the ADMs. Since the footprint sizes are different between CERES-Aqua and CERES-NPP, will using ADMs developed based upon CERES-Aqua measurements for CERES-NPP flux inversion introduce any uncertainties in the CERES-NPP flux? Additionally, ADMs are scene type dependent, it is important to use consistent scene identification for developing and applying the ADMs. However, the VIIRS channels are not identical to those of MODIS, especially the lack of 6.7 $\mu$m and 13.3 $\mu$m channels, caused the cloud properties retrieved from MODIS and VIIRS differ from each other. These differences affect the scene identification used to select the ADMs for flux inversion and thus can lead to additional uncertainties in the CERES-NPP flux. In this study, we design a simulation study to quantify the CERES-NPP flux uncertainties due to the footprint size difference alone, and due to both the footprint size and cloud property differences.

## 2. Comparison between CERES-Aqua and CERES-NPP

Besides the altitude differences between Aqua and NPP satellites, they are also different in other orbital characteristics. For example, the orbital period for Aqua is about 98.82 minutes, while it is about 101.44 minutes for NPP; and the orbital inclination for Aqua is about 98.20°, while it is about 98.75° for NPP. These orbital differences result in different local overpass times between Aqua and NPP and their orbits fly over each other about every 64 hours. These simultaneous observations from Aqua and NPP are matched to compare SW and LW radiances using CERES Aqua Edition 4 Single Scanner Footprint TOA/Surface Fluxes and Clouds (SSF) product and CERES NPP Edition 1 SSF product. Here we use $I_a^m$ to denote the CERES-Aqua (subscript $a$) measured (superscript $m$) radiance, and $I_n^m$ as the CERES-NPP (subscript $n$) measured radiance. Similarly, $F_a^m$ and $F_n^m$ are the fluxes derived from $I_a^m$ and $I_n^m$ using CERES Aqua ADMs. The matching criteria used for SW radiances are that the latitude and longitude differences between the Aqua footprints and the NPP footprints are less than 0.05 degree, solar zenith angle and viewing zenith angle differences

are less than 2 degrees, and relative azimuth angle difference is less than 5 degrees. The matching criteria used here also provide a tight constraint on scattering angles, with about 95.6% and 99.9% of the matched footprints having scattering angle differences less than 2 degrees and 3 degrees, respectively. Same latitude and longitude matching criteria are used for LW radiances and the viewing zenith angle difference between the Aqua footprints and the NPP footprints is less than 2 degrees.

Figure 1 shows the SW, daytime LW, and nighttime LW radiance comparisons between CERES-Aqua and CERES-NPP using matched footprints of 2013 and 2014. The total number of matched footprints, the mean $I_a^m$ and $I_n^m$, and the root-mean-square (RMS) errors are summarized in Table 1. The mean SW $I_n^m$ is about 1 $\mathrm{Wm^{-2}sr^{-1}}$ greater than $I_a^m$, the mean daytime LW $I_n^m$ is about 0.4 $\mathrm{Wm^{-2}sr^{-1}}$ smaller than $I_a^m$, and the nighttime LW $I_n^m$ and $I_a^m$ agree to within 0.1 $\mathrm{Wm^{-2}sr^{-1}}$. Excluding matched footprints with scattering angle difference greater than 2 degrees does not change the SW comparison result. These comparisons include data taken from nadir to oblique viewing angles ($\theta > 60$). The RMS errors remain almost the same when we compare the radiances taken at different $\theta$ ranges. Footprint size differences may also contribute to the radiance differences, but these radiance differences should be random. It is likely that the footprint size differences can increase the RMS errors, but the mean radiance differences are mostly resulted from calibration differences between CERES-Aqua and CERES-NPP. As mentioned earlier, the daytime CERES LW radiance is derived as the difference between total channel and SW channel measurements, and the nighttime CERES LW radiance is directly derived from the total channel measurements. The differences shown in Table 1 indicate that the agreement of the total channels between CERES-Aqua and CERES-NPP are better than that of the SW channels, leading to a smaller daytime LW difference than SW difference. Loeb et al. (2016) examined the normalized instrument gains for the total and SW channels for CERES FM1-FM5 since the beginning of the mission (BOM). The total channel response to LW radiation has gradually increased with time for all instruments. For the two instruments (FM3 and FM5) that are of interest

here, the increases relative to the BOM are 0.7% for FM3 and 0.4% for FM5. The SW channel response increases about 0.4% for FM3 and decreases by 0.2% for FM5. Exact causes for the calibration differences between CERES-Aqua and CERES-NPP are not yet known and more research are needed to understand their differences. The future plan is to place CERES-NPP on the same radiometric scale as CERES-Aqua.

Flux comparison using the same matched footprints are shown in Figure 2 and the mean $F_a^m$ and $F_n^m$, and the RMS errors between them are summarized in Table 1. Consistent with the radiance comparisons, the mean SW $F_n^m$ is about 3.8 Wm$^{-2}$ greater than $F_a^m$, the mean daytime LW $F_n^m$ is about 1.0 Wm$^{-2}$ smaller than $F_a^m$, and the mean nighttime LW $F_n^m$ is about 0.3 Wm$^{-2}$ smaller than $F_a^m$. When we compare the relative RMS errors (RMS error divided by the mean Aqua value) between radiance and flux, the relative flux RMS errors (6.4% for SW, 2.2% for daytime LW, and 1.4% for nighttime LW) are always slightly larger than the relative radiance RMS errors (6.0% for SW, 2.1% for daytime LW, and 1.1% for nighttime LW). This indicates that additional uncertainties are added when the radiances are converted to fluxes.

However, we cannot directly compare the gridded monthly mean fluxes from Aqua and NPP as their overpass times differ. Figure 3 shows the monthly mean TOA insolation difference between CERES-NPP and CERES-Aqua for April 2013. Insolation for NPP overpass times is greater than that for Aqua overpass times over most regions, except over the northern high latitude where NPP has significantly more overpasses at $\theta_0 > 70°$ than Aqua. Regional differences as large as 30 Wm$^{-2}$ are observed over the tropical regions and north of 60°N. Globally, the CERES-NPP monthly mean insolation is greater than that of CERES-Aqua by 13.4 Wm$^{-2}$ for this month. When we compare the monthly gridded TOA reflected SW flux between CERES-NPP and CERES-Aqua (Figure 4a), the difference features in high latitude regions (north of 60°N and south of 60°S) resemble those of the insolation differences. We then compare the albedo between CERES-NPP and CERES-Aqua (Figure 4b). Over most regions, the albedo from CERES-NPP is greater than that from CERES-Aqua, except over

parts of tropical oceans and Antarctica where some negative differences are observed. The global monthly mean albedo from CERES-NPP is greater than that from CERES-Aqua by 0.003 (1.02%). The albedo difference is mostly from the calibration differences (see Figure 1a and Table 1) , while the footprint size difference and scene identification difference also contribute to the albedo difference.

The CERES cloud working group developed sophisticated cloud detection algorithms using visible and infrared channels of MODIS separately for polar and non-polar regions and for daytime, twilight, and nighttime (Trepte et al. 2010). However, these detection algorithms have to be modified to be applicable to the VIIRS observations (Qing Trepte, personal communication), as some of the MODIS channels utilized for cloud detection are not available on VIIRS. These modifications include replacing the 2.1 $\mu$m MODIS channel with the 1.6 $\mu$m VIIRS channel, and replacing detection tests using MODIS 6.7 $\mu$m and 13.3 $\mu$m channels with VIIRS 3.7 $\mu$m and 11 $\mu$m channels, and supplement with tests utilizing VIIRS 1.6 $\mu$m channel and the brightness temperature differences between 11 $\mu$m and 12 $\mu$m. These changes mainly affect cloud detections over the polar regions. The parameterization of 1.24 $\mu$m reflectance was regenerated for VIIRS using improved wavelength and insolation weighting, which affects cloud optical depth retrieval over the snow/ice surfaces (Szedung Sun-Mack, personal communication). These changes result in different cloud properties retrieved using MODIS and VIIRS, especially over the polar regions. Figure 5 shows the daytime cloud fraction and cloud optical depth difference between VIIRS and Aqua-MODIS for April 2013. Cloud fraction retrieved from VIIRS is greater than that from MODIS by up to 10% and cloud optical depth from VIIRS is smaller than that from MODIS by 2∼3 over part of the Antarctic. Cloud fraction from VIIRS over the northern high-latitude snow regions is smaller than that from MODIS, while the optical depth from VIIRS is greater than that from MODIS. Over the Arctic, cloud optical depth from VIIRS is much larger than that from MODIS. Over the ocean between 60°S and 60°N, the differences in cloud fraction seem rather random while the differences in cloud optical depth is mostly positive

<sub>225</sub> (VIIRS retrieval is greater than Aqua-MODIS retrieval).

<sub>226</sub> Given that the footprint sizes and overpass times are different between CERES-Aqua
<sub>227</sub> and CERES-NPP, in addition to the calibration differences and cloud retrieval differences
<sub>228</sub> between them, fluxes from these CERES instruments cannot be compared directly to assess
<sub>229</sub> the effects of footprint size difference and cloud property difference on flux uncertainty.

# <sub>230</sub> 3.  Method

<sub>231</sub> To quantify the footprint size and cloud retrieval effect on flux inversion without having
<sub>232</sub> to account for the calibration and overpass time differences, we design a simulation study
<sub>233</sub> using the MODIS pixel level data and the Aqua-Earth-Sun geometry. MODIS spectral
<sub>234</sub> measurements are used to retrieve cloud properties and aerosol optical depth. These pixel-
<sub>235</sub> level imager-derived aerosol and cloud properties, and spectral narrowband (NB) radiances
<sub>236</sub> from MODIS are convolved with the CERES PSF to provide the most accurate aerosol and
<sub>237</sub> cloud properties that are spatially and temporally matched with the CERES broadband
<sub>238</sub> radiance data. Figure 6 illustrates the process of generating the simulated CERES-Aqua
<sub>239</sub> and CERES-NPP footprints from the MODIS pixels. We first use the CERES-Aqua PSF
<sub>240</sub> to convolve the aerosol/cloud properties, and the MODIS NB radiances (and other ancillary
<sub>241</sub> data) into Aqua-size footprints (left portion of Figure 6), as is done for the standard CERES-
<sub>242</sub> Aqua SSF product. These NB radiances for the simulated CERES-Aqua footprints are
<sub>243</sub> denoted as $I_a^s(\lambda)$, where superscript 's' is for the simulated (in contract to superscript 'm'
<sub>244</sub> for the measured). We then increase the footprint size to be that of NPP and use the
<sub>245</sub> CERES-NPP PSF to average the MODIS NB radiances, cloud/aerosol properties, and other
<sub>246</sub> ancillary data into the simulated NPP footprints. NB radiances for the simulated CERES-
<sub>247</sub> NPP footprints are denoted as $I_n^s(\lambda)$.

<sub>248</sub> Four months (July 2012, October 2012, January 2013, and April 2013) of simulated
<sub>249</sub> CERES-Aqua and CERES-NPP data were created. For every CERES-Aqua footprint, it

contains the broadband SW and LW radiances measured by the CERES instrument. The simulated NPP footprints, however, do not contain broadband radiances. To circumvent this issue, we developed narrowband-to-broadband coefficients to convert the MODIS NB radiances to broadband radiances.

The Edition 4 CERES-Aqua SSF data from July 2002 to September 2007 are used to derive the narrowband-to-broadband (NB2BB) regression coefficients separately for SW, daytime LW, and nighttime LW. Seven MODIS spectral bands (0.47, 0.55, 0.65, 0.86, 1.24, 2.13, and 3.7 $\mu$m) are used to derive the broadband SW radiances, and the SW regression coefficients are calculated for every calendar month for discrete intervals of solar zenith angle, viewing zenith angle, relative azimuth angle, surface type, snow/non-snow conditions, cloud fraction, and cloud optical depth. Five MODIS spectral bands (6.7, 8.5, 11.0, 12.0, and 14.2 $\mu$m) are used to derive the broadband LW radiances, and the LW regression coefficients are calculated for every calendar month for discrete intervals of viewing zenith angle, precipitable water, surface type, snow/none-snow conditions, cloud fraction, and cloud optical depth. The 20 IGBP surface types are grouped into 8 surface types: ocean, forest, savanna, grassland, dark desert, bright desert, the Greenland permanent snow, and the Antarctic permanent snow. When there is sea ice over the ocean and snow over the land surface types, regression coefficients for ice and snow conditions are developed (only footprints with 100% sea ice/snow coverage are considered).

These SW and LW NB2BB regression coefficients are then applied to $I_a^s(\lambda)$ and $I_n^s(\lambda)$ to derive the broadband radiances, $I_a^s$ and $I_n^s$, for simulated footprints of CERES-Aqua and CERES-NPP, shown on the left and right of Figure 6, if the footprint consists of a single surface type. As both simulated CERES-Aqua and CERES-NPP footprints use the Aqua-Earth-Sun geometry, $I_a^s$ and $I_n^s$ have the same Sun-viewing geometry. Even though the CERES-Aqua footprints contained the broadband radiances from CERES observations ($I_a^m$), we choose to use the broadband radiances calculated using the NB2BB regressions to ensure that $I_a^s$ and $I_n^s$ are consistently derived. Doing so we can isolate the flux differences between

<sub>277</sub> simulated CERES-Aqua and simulated CERES-NPP caused by footprint size difference.

<sub>278</sub>    The cloud properties in the simulated CERES-Aqua footprints and in the simulated
<sub>279</sub> CERES-NPP footprints are all based upon the MODIS retrievals, so the scene identifica-
<sub>280</sub> tions used to select ADMs for flux inversion are almost the same for both the simulated
<sub>281</sub> CERES-Aqua and the CERES-NPP, except for small differences due to differing footprint
<sub>282</sub> sizes. As demonstrated in Figure 5, cloud properties differ between the MODIS and the VI-
<sub>283</sub> IRS retrievals. These cloud retrieval differences affect the anisotropy factors selected for flux
<sub>284</sub> inversion. To simulate both the footprint size and cloud property differences, cloud fraction
<sub>285</sub> and cloud optical depth retrievals from MODIS convolved in the simulated CERES-NPP
<sub>286</sub> footprints are adjusted to be similar to those from VIIRS retrievals to assess how cloud
<sub>287</sub> retrieval differences affect the flux. To accomplish this, daily cloud fraction ratios of VIIRS
<sub>288</sub> to MODIS are calculated for each $1°$ latitude by $1°$ longitude grid box. These ratios are then
<sub>289</sub> applied to the cloudy footprints of MODIS retrieval to adjust the MODIS cloud fractions
<sub>290</sub> to be nearly the same as those from VIIRS retrieval. Note no adjustment is done for clear
<sub>291</sub> footprints. Similarly, daily cloud optical depth ratios of VIIRS to MODIS are calculated us-
<sub>292</sub> ing cloudy footprints for each $1°$ by $1°$ grid box. These ratios are used to adjust the MODIS
<sub>293</sub> retrieved cloud optical depth to be close to those from VIIRS retrievals. The process of gen-
<sub>294</sub> erating the simulated CERES-NPP footprints with VIIRS-like cloud retrievals is illustrated
<sub>295</sub> on the lower right portion of Figure 6.

<sub>296</sub>    Aqua ADMs are then used to convert $I_a^s$ and $I_n^s$ to fluxes, $F_a^s$ and $F_n^s$, for the simulated
<sub>297</sub> CERES-Aqua and CERES-NPP footprints using the cloud properties retrieved from MODIS
<sub>298</sub> observations for scene type identification. To further access the effects of both footprint size
<sub>299</sub> and cloud property differences on flux inversion, Aqua ADMs are used to convert $I_n^s$ to flux,
<sub>300</sub> $F_n'^s$, for the simulated CERES-NPP footprints using VIIRS-like cloud properties for scene
<sub>301</sub> identification.

# 4. Results

We first compare the footprint-level fluxes between simulated CERES-Aqua and simulated CERES-NPP using data of April 1, 2013 (about 700,000 footprints). As the cloud fraction and cloud optical depth adjustments are done at the grid box level, it is not feasible to compare footprint-level $F_a^s$ and $F_n^{'s}$, and only footprint-level $F_a^s$ and $F_n^s$ are compared. For SW, the bias between $F_a^s$ and $F_n^s$ is 0.1 $\mathrm{Wm}^{-2}$ and the RMS error is 4.7 $\mathrm{Wm}^{-2}$. For LW, the biases is close to zero and the RMS errors are 1.3 $\mathrm{Wm}^{-2}$ and 0.9 $\mathrm{Wm}^{-2}$ for daytime and nighttime, respectively. These flux RMS errors are much smaller than those listed in Table 1, indicating that calibration differences are responsible for most of the flux differences between CERES-Aqua and CERES-NPP measurements. However, we should avoid direct comparisons between these two sets of RMS errors, as they are derived using different time period.

We now compare the monthly grid box (1° latitude by 1° longitude) mean fluxes from the three simulations outlined in the previous section. Differences between $F_n^s$ and $F_a^s$ are used to assess the CERES-NPP gridded monthly mean instantaneous flux uncertainties due to the footprint size difference, and differences between $F_n^{'s}$ and $F_a^s$ are used to assess the CERES-NPP gridded monthly mean instantaneous flux uncertainties due to both the footprint size and cloud property differences.

The monthly mean instantaneous TOA SW fluxes for simulated CERES-Aqua ($F_a^s$) are shown in Figure 7(a) for April 2013. Note these fluxes are different from those in the Edition 4 Aqua SSF product as the CERES measured radiances differ from those inferred using NB2BB regression coefficients. The flux differences caused by the footprint size difference between the simulated CERES-NPP and the simulated CERES-Aqua ($F_n^s - F_a^s$) are shown in Figure 7(b). Grid boxes in white indicate that the number of footprints with valid SW fluxes differ by more than 2% between simulated CERES-Aqua and CERES-NPP, as the NB2BB regressions are only applied to footprints that are consist of the same surface types which result in fewer footprints with valid fluxes for CERES-NPP than for CERES-Aqua. The

footprint size difference between CERES-Aqua and CERES-NPP introduces an uncertainty that rarely exceeds 4.0 $\text{Wm}^{-2}$ in monthly gridded CERES-NPP instantaneous SW fluxes. For global monthly mean instantaneous SW flux, the simulated CERES-NPP has a low bias of 0.4 $\text{Wm}^{-2}$ compares to the simulated CERES-Aqua, and the RMS error between them is 2.4 $\text{Wm}^{-2}$. Results from the other three months are very similar to April 2013 (not shown).

Figure 7(c) shows the SW flux difference caused by both the footprint size and cloud property differences ($F_n^{'s} - F_a^s$). Adding the cloud property differences increase the CERES-NPP flux uncertainty comparing to when only footprint size differences are considered (Figure 7(b)), monthly gridded instantaneous flux uncertainty over the Arctic ocean can exceed 20 $\text{Wm}^{-2}$. Accounting for cloud property differences, the global monthly mean instantaneous SW flux from simulated CERES-NPP has a high bias of 1.1 $\text{Wm}^{-2}$ and the RMS error is increased to 5.2 $\text{Wm}^{-2}$. Over the Arctic Ocean, the cloud optical depth from VIIRS retrieval is much greater than that from the MODIS retrieval while the difference in cloud fraction is relatively small. Anisotropic factors for thick clouds are smaller than those for thin clouds at oblique viewing angles, and are larger for near-nadir viewing angles. The viewing geometries over the Arctic Ocean produced more smaller anisotropic factors than larger ones when MODIS cloud optical depths were replaced with VIIRS-like cloud optical depths, which resulted in larger fluxes when using VIIRS-like cloud properties for flux inversion.

The daytime and nighttime LW flux from the simulated CERES-Aqua footprints, LW flux differences due to footprint size difference, and LW flux difference due to both footprint size difference and cloud property difference are shown in Figures 8 and 9. The effect of footprint size on gridded monthly mean daytime and nighttime LW flux is generally within 1.0 $\text{Wm}^{-2}$. For global monthly mean LW flux, the differences between $F_n^s - F_a^s$ are close to zero, and the RMS errors between them are about 0.8 $\text{Wm}^{-2}$ and 0.2 $\text{Wm}^{-2}$ for daytime and nighttime LW fluxes. When cloud property differences are also considered, their effect on gridded monthly mean LW fluxes increases to about 2 $\text{Wm}^{-2}$. The RMS errors of global monthly mean LW flux increase slightly to about 0.9 $\text{Wm}^{-2}$ and 0.5 $\text{Wm}^{-2}$ for daytime and

nighttime. The LW fluxes showed much less sensitivity to cloud property changes than the SW fluxes, especially over the Arctic Ocean where cloud optical depth changed significantly. This is because the LW ADMs over the snow/ice surfaces have very little sensitivity to cloud optical depth (Su et al. 2015a), but they were developed for discrete cloud fraction intervals and larger flux changes are noted in regions experiencing large cloud fraction changes.

# 5. Summary and discussion

The scene-type dependent ADMs are used to convert the radiances measured by the CERES instruments to fluxes. Specific empirical ADMs were developed for CERES instruments on TRMM, Terra, and Aqua (Loeb et al. 2003, 2005; Su et al. 2015a). As there is only one CERES instrument on NPP and it has been placed in cross-track mode since launch, it is not possible to construct a set of ADMs specific for CERES on NPP. Edition 4 Aqua ADMs (Su et al. 2015a) are thus used for flux inversions for CERES-NPP measurements. However, the altitude of the NPP orbit is higher than that of the Aqua orbit resulting in a larger CERES footprint size on NPP than on Aqua. Given that the footprint size of CERES-NPP is different from that of CERES-Aqua, we need to quantify the CERES-NPP flux uncertainty caused by using the CERES-Aqua ADMs. Furthermore, there are some differences between the imagers that are on the same spacecrafts as CERES-Aqua (MODIS) and CERES-NPP (VIIRS), as VIIRS lacks the 6.7 $\mu$m and 13.3 $\mu$m channels. These spectral differences and algorithm differences lead to notable cloud fraction and cloud optical depth differences retrieved from MODIS and VIIRS. As the anisotropy factors are scene-type dependent, differences in cloud properties will also introduce uncertainties in flux inversion. Furthermore, the calibrations between CERES instruments on Aqua and on NPP also are different from each other. Comparisons using two years of collocated CERES-Aqua and CERES-NPP footprints indicate that the SW radiances from CERES-NPP are about 1.5% greater than those from CERES-Aqua, the daytime LW radiances from CERES-NPP are

about 0.5% smaller than those from CERES-Aqua, and the nighttime LW radiances agree to within 0.1%.

To quantify the flux uncertainties due to the footprint size difference between CERES-Aqua and CERES-NPP, and due to both the footprint size difference and cloud property difference, we use the MODIS pixel level data to simulate the CERES-Aqua and CERES-NPP footprints. The simulation is designed to isolate the effects of footprint size difference and cloud property difference on flux uncertainty from calibration difference between CERES-NPP and CERES-Aqua. The pixel-level MODIS spectral radiances, the imager-derived aerosol and cloud properties, and other ancillary data are first convolved with the CERES Aqua PSF to generate the simulated CERES-Aqua footprints, and then convolved with the CERES NPP PSF to generate the simulated CERES-NPP footprints. Broadband radiances within the simulated CERES-Aqua and CERES-NPP footprints are derived using the MODIS spectral bands based upon narrowband-to-broadband regression coefficients developed using five years of Aqua data to ensure consistency between broadband radiances from simulated CERES-Aqua and CERES-NPP. These radiances are then converted to fluxes using the CERES-Aqua ADMs. The footprint size difference between CERES-Aqua and CERES-NPP introduces instantaneous flux uncertainties in monthly gridded CERES-NPP of less than $4.0\,\mathrm{Wm^{-2}}$ for SW, and less than $1.0\,\mathrm{Wm^{-2}}$ for both daytime and nighttime LW. The global monthly mean instantaneous SW flux from simulated CERES-NPP has a low bias of $0.4\,\mathrm{Wm^{-2}}$ compares to that from simulated CERES-Aqua, and the RMS error between them is $2.4\,\mathrm{Wm^{-2}}$. The biases in global monthly mean LW fluxes are close to zero, and the RMS errors between simulated CERES-NPP and simulated CERES-Aqua are about $0.8\,\mathrm{Wm^{-2}}$ and $0.2\,\mathrm{Wm^{-2}}$ for daytime and nighttime global monthly mean LW fluxes.

The cloud properties in the simulated CERES-Aqua footprints and in the simulated CERES-NPP footprints are all based upon MODIS retrievals, but in reality cloud properties retrieved from VIIRS differ from those from MODIS. To assess the flux uncertainty from scene identification differences, cloud fraction and cloud optical depth in the simulated

CERES-NPP footprints are perturbed to be more like the VIIRS retrievals. When both footprint size and cloud property differences are considered, the uncertainties of monthly gridded CERES-NPP SW flux can be up to 20 $Wm^{-2}$ in the Arctic regions where cloud optical depth retrievals from VIIRS differ significantly from MODIS. The global monthly mean instantaneous SW flux from simulated CERES-NPP has a high bias of 1.1 $Wm^{-2}$ and the RMS error is increased to 5.2 $Wm^{-2}$. LW flux shows less sensitivity to cloud property differences than SW flux, with the uncertainties of about 2.0 $Wm^{-2}$ in monthly gridded LW flux, and the RMS errors increases to 0.9 $Wm^{-2}$ and 0.5 $Wm^{-2}$ for daytime and nighttime LW flux.

Su et al. (2015b) quantified the global monthly 24hr-averaged flux uncertainties due to CERES ADMs using direct integration tests, and concluded that the RMS errors are less than 1.1 $Wm^{-2}$ and 0.8 $Wm^{-2}$ for 24hr-averaged TOA SW and LW fluxes. The uncertainty for global monthly instantaneous SW flux is approximately twice the uncertainty of 24hr-averaged flux. This simulation study indicates that the footprint size differences between CERES-NPP and CERES-Aqua introduce flux uncertainties that are within the uncertainties of the CERES ADMs. However, the uncertainty assessment provided here should be considered as the low end, as many regions (especially over land, snow, and ice) were not included due to sample number differences within the grid boxes. When cloud property differences are accounted for, the SW flux uncertainties increase significantly and exceed the uncertainties of the CERES ADMs. These findings indicate that inverting CERES-NPP flux using CERES-Aqua ADMs resulting in flux uncertainties that are within the ADMs uncertainties as long as the cloud retrievals between VIIRS and MODIS are consistent. When the cloud retrieval differences between VIIRS and MODIS are accounted for, the SW flux uncertainties exceed those of the CERES ADMs. To maintain the consistency of the CERES climate data record, it is thus important to develop cloud retrieval algorithms that account for the capabilities of both MODIS and VIIRS to ensure consistent cloud properties from both imagers.

435

*Acknowledgments.*

This research was supported by the NASA CERES project. The CERES data were obtained from the NASA Langley Atmospheric Science Data Center at https://eosweb.larc.nasa.gov/project/ceres/ssf_table. We thank Norman Loeb, Szedung Sun-Mack, Qing Trepte, and Patrick Minnis for helpful discussions, and the three reviewers for their constructive comments and suggestions which have significantly improved this paper.

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

# List of Tables

TABLE 1. Comparison of CERES-Aqua and CERES-NPP measured SW, daytime LW, and nighttime LW radiances ($Wm^{-2}sr^{-1}$) and fluxes ($Wm^{-2}$) using matched footprints of 2013 and 2014.

|  | SW | Daytime LW | Nighttime LW |
| --- | --- | --- | --- |
| Sample Number | 147894 | 192178 | 187880 |
| Mean CERES-Aqua Radiance | 68.1 | 77.4 | 74.4 |
| Mean CERES-NPP Radiance | 69.2 | 77.0 | 74.3 |
| Radiance RMS Error | 4.1 | 1.6 | 0.8 |
| Mean CERES-Aqua Flux | 230.1 | 235.7 | 226.4 |
| Mean CERES-NPP Flux | 233.9 | 234.7 | 226.1 |
| Flux RMS Error | 14.6 | 5.0 | 3.1 |

# List of Figures

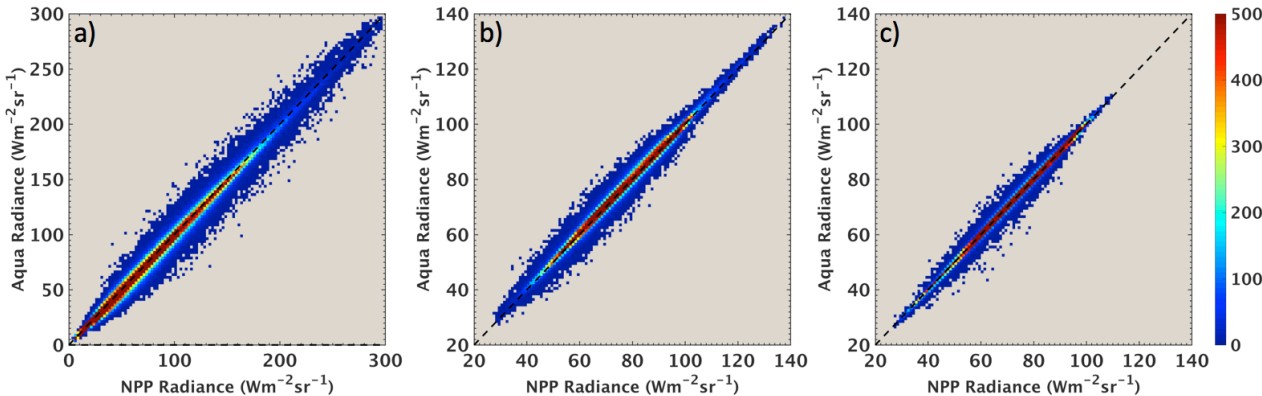

FIG. 1. Radiance comparisons between matched CERES-Aqua and CERES-NPP footprints, (a) SW; (b) daytime LW; and (c) nighttime LW using data of 2013 and 2014. The total number of footprints, the mean radiances, and the radiance RMS errors are summarized in Table 1.

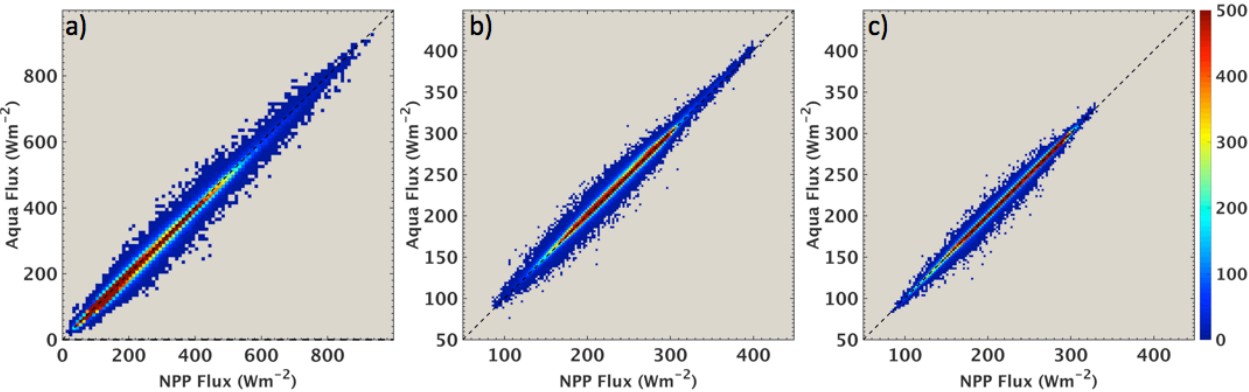

Fig. 2. Flux comparisons between matched CERES-Aqua and CERES-NPP footprints, (a) SW; (b) daytime LW; and (c) nighttime LW using data of 2013 and 2014. The total number of footprints, the mean fluxes, and the flux RMS errors are summarized in Table 1.

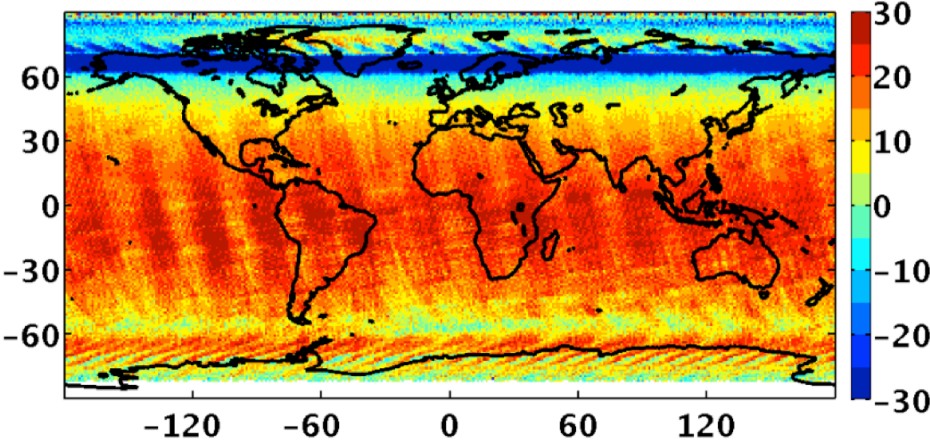

Fɪɢ. 3. Monthly mean insolation difference (Wm$^{-2}$) between CERES-NPP and CERES-Aqua (NPP-Aqua) for April 2013.

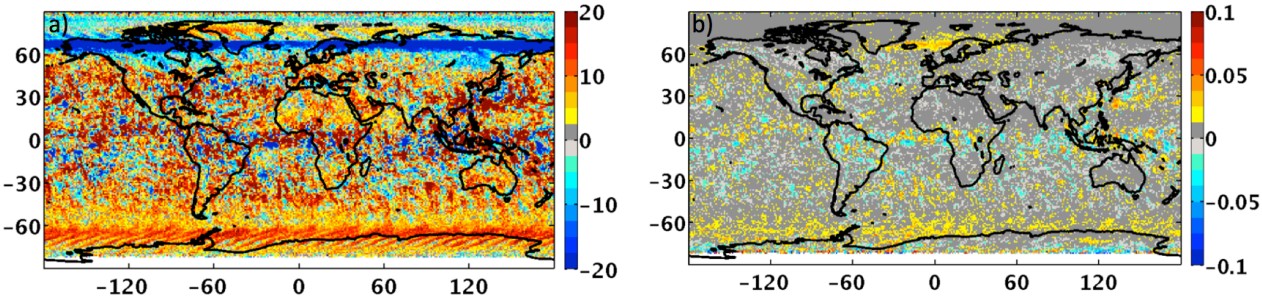

FIG. 4. Monthly mean a) TOA reflected SW flux difference between CERES-NPP and CERES-Aqua (NPP-Aqua), and b) albedo difference between CERES-NPP and CERES-Aqua (NPP-Aqua) for April 2013.

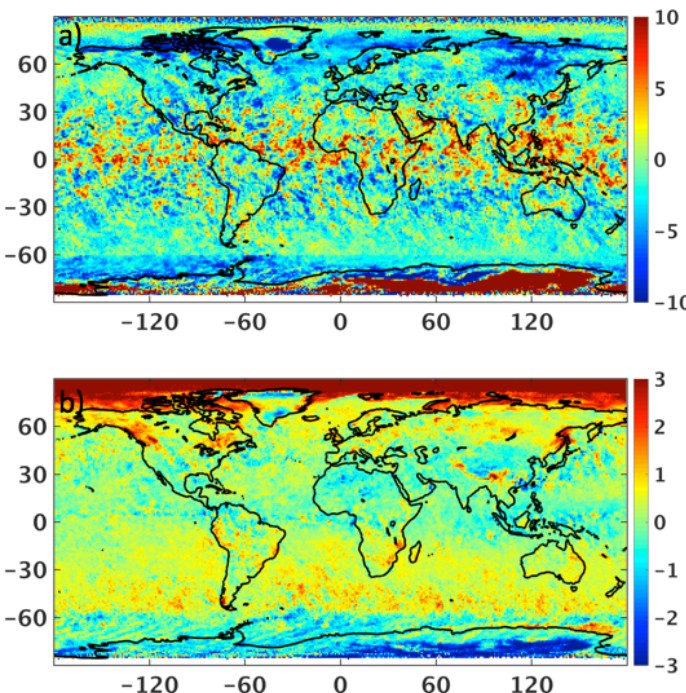

Fɪɢ. 5. Cloud fraction (a) and cloud optical depth (b) differences between VIIRS and MODIS (VIIRS-MODIS) retrievals for April 2013.

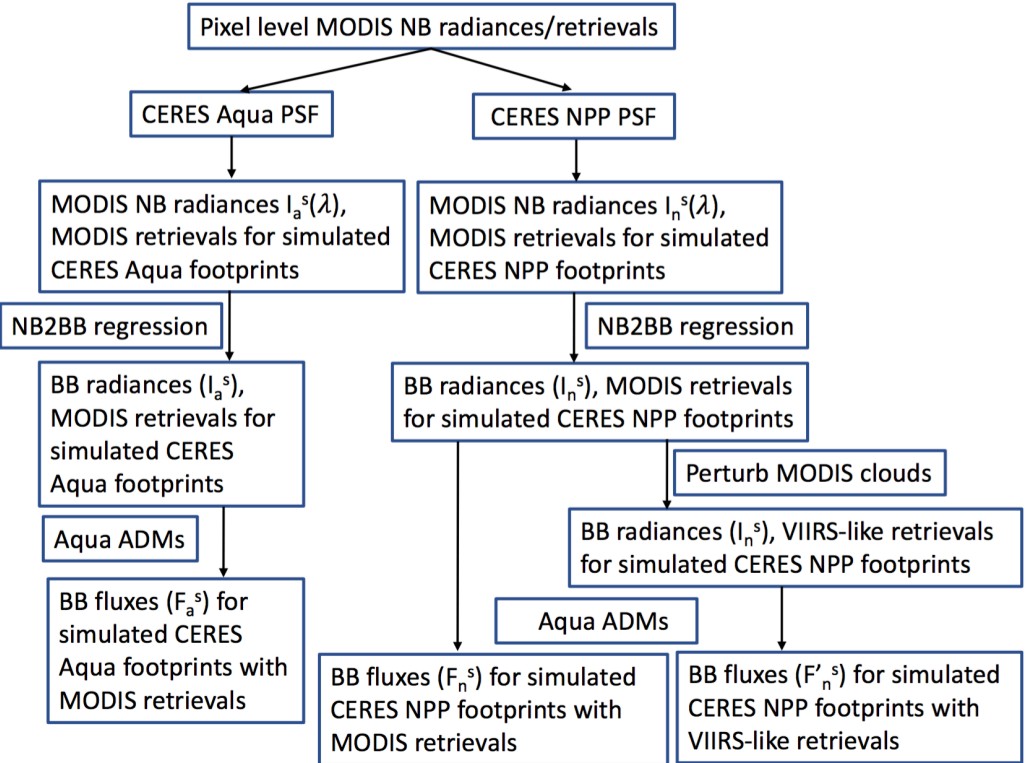

FIG. 6. Schematic diagram of convoluting the MODIS pixels into the simulated Aqua and NPP footprints. Left depicts the processes involved in producing the simulated Aqua footprints; middle for simulated NPP footprints with MODIS retrievals; and right for simulated NPP footprints with VIIRS-like retrievals.

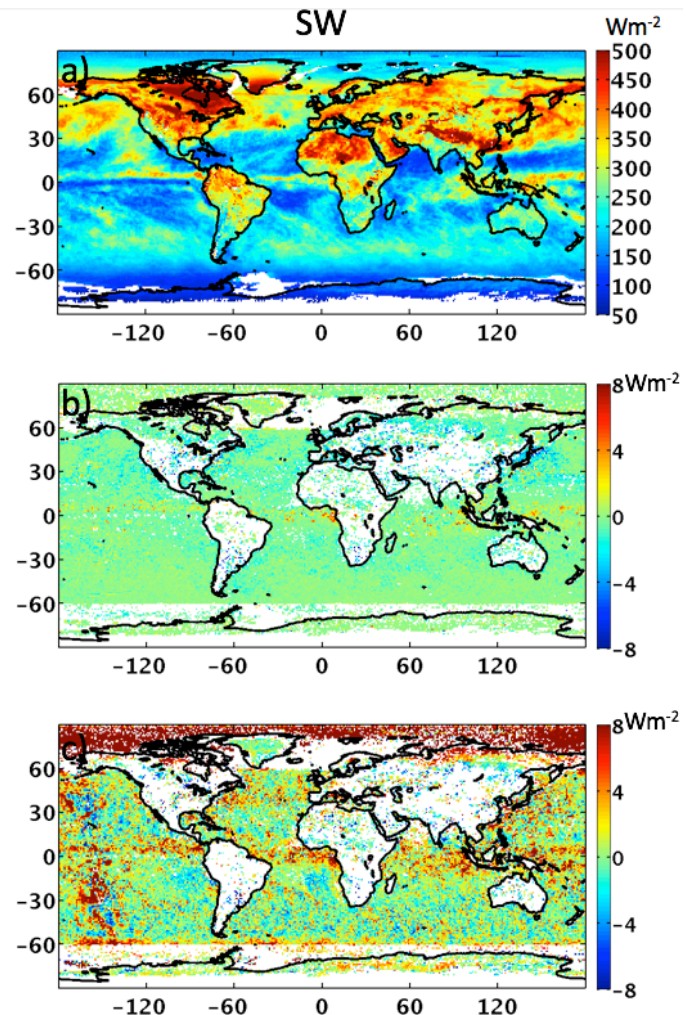

FIG. 7. The gridded monthly mean TOA instantaneous SW fluxes from the simulated Aqua footprints ($F_a^s$, a), the flux differences caused by footprint size difference between simulated NPP and simulated Aqua ($F_n^s - F_a^s$, b), and the flux differences caused by both footprint size and cloud property differences ($F_n'^s - F_a^s$, c) using April 2013 data. Regions shown in white have large sample number differences between simulated Aqua and simulated NPP.

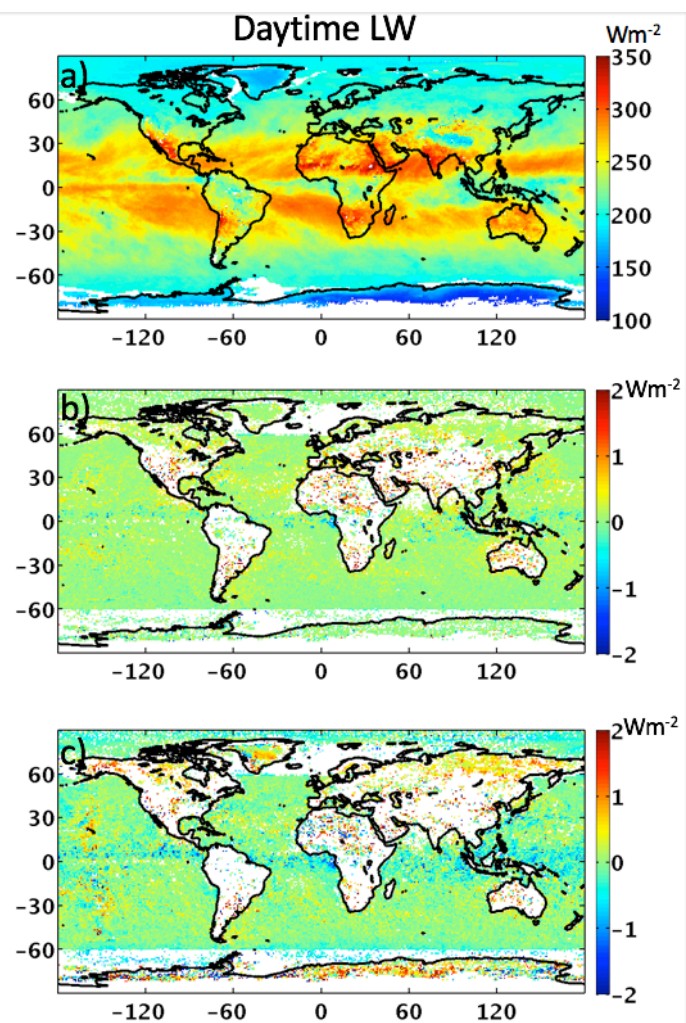

Fɪɢ. 8. The gridded monthly mean TOA daytime LW fluxes from the simulated Aqua footprints ($F_a^s$, a), the flux differences caused by footprint size difference between simulated NPP and simulated Aqua ($F_n^s - F_a^s$, b), and the flux differences caused by both footprint size and cloud property differences ($F_n^{'s} - F_a^s$, c) using April 2013 data. Regions shown in white have large sample number differences between simulated Aqua and simulated NPP.

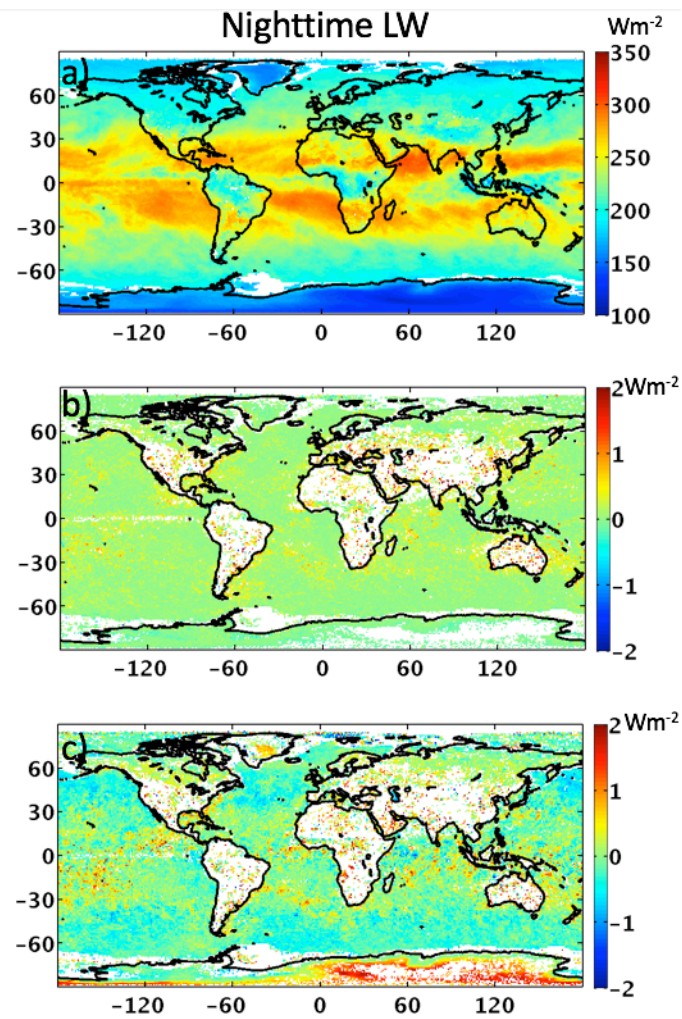

FIG. 9. The gridded monthly mean TOA nighttime LW fluxes from the simulated Aqua footprints ($F_a^s$, a), the flux differences caused by footprint size difference between simulated NPP and simulated Aqua ($F_n^s - F_a^s$, b), and the flux differences caused by both footprint size and cloud property differences ($F_n^{'s} - F_a^s$, c) using April 2013 data. Regions shown in white have large sample number differences between simulated Aqua and simulated NPP.