# Peer review of "The effects of different footprint sizes and cloud algorithms on the"

_Atmospheric Measurement Techniques, 2017_

## Referee Comment (RC1) · Anonymous Referee #3 · 11 May 2017

Peer review for Su et al paper.

Overall impression: Publish pending very major revisions.

General comments:

1. Whereas the overall research is of great importance as the various EOS algorithms are moving into the JPSS era, this paper is plain and simple incomplete. The authors give a pretty good discussion of the footprint size difference estimation experiments, in the title and the abstract they speak of retrieval algorithm differences as well, yet in the text the treatment of retrieval algorithm barely gets a sentence or two. I feel like the entire discussion of the retrieval algorithm is missing. In research it is important for published result to be reproducible to at least some degree. This work is definitely not reproducible. The footprint difference evaluation needs more detail and the algorithm difference evaluation plain and simple needs to be written.

2. Throughout, please consider the audience may not know what your different acronyms are. Please expand in text or provide an acronym table.

Specific comments:

Line 7: CERES, first mention of the instrument, please expand
Line 12: same thing for VIIRS and MODIS, consider the audience may not know what the acronyms are.
Line 26: "cloud property differences", elaborate what exact cloud properties you mean
Line 48: "flied" is not a word
Line 49: "were launch" is grammatically incorrect
Line 56-57: spacecrafts is not a word
Lines 115-117: why have you not considered the scattering angle?
Line 128: "the nighttime LW radiance is simply from the total channel measurements" doesn't make any sense.
Line 157: snow /no snow conditions, how are you getting this information?
Line 159: "to derived" is grammatically incorrect
Line 163-166: how are you getting snow/no snow information?
Line 178: replace 'assessed' with 'ascertained'

You talk about the simulated footprint generation, but where is the discussion of the algorithm differences? "We tweaked it" is not good enough.

Line 192: "that consist with" is not English, "result in less footprints" is not either.
Line 204-210: Why? Please explain
Line 224-227: This does not make any sense to anyone who might theoretically even consider checking your results.
Line 235: Why did your flux increase? Explain
Line 273: replace 'is' with 'are'

Line 275: replace 'these' with 'that'
Line 282: replace 'thus' with 'to'

Comments on figures:

Figure 2: so these CERES footprints are misaligned like that in real life or just in illustration? Please clarify.

Figures 3-6: can't be evaluated because I don't know what I'm looking at. There is no explanation in the text as to why the images look they way they do. Could be a thousand different reasons, so figures do not make sense.

---

## Referee Comment (RC2) · Anonymous Referee #4 · 13 Jun 2017

Review of manuscript amt-2017-75

"The effects of different footprint sizes and cloud algorithms on the top-of-atmosphere radiative flux calculation from CERES instrument on Suomi-NPP" by Su et al.

Major comments

This manuscript attempts to quantify the flux uncertainties in the NPP-CERES fluxes because NPP does not have a rotating CERES instrument to generate NPP specific ADM as the TERRA and AQUA satellites had and has to borrow the ADM generated from AQUA satellite. The study uses AQUA MODIS radiances to create simulated fluxes for both AQUA and NPP footprints – going through the same narrowband to broadband conversion and using the same ADM for radiance to flux conversion to isolate the differences due to footprint size alone. The differences due to both footprint size and cloud properties are examined with VIIRS-adjusted cloud properties and subsequent ADM. This study helps to understand and quantify the uncertainty in the current NPP CERES fluxes due to missing its own ADM. However, the description of the methodology could be much improved by adding proper flow chart and naming convention. The other missing piece of information is how the uncertainties estimated with simulated fluxes starting from MODIS pixel radiances are measured against actual retrieved NPP CERES flux? Similar comparisons between the matched NPP and AQUA radiances (Figure 1) could be applied to match fluxes. Likewise, gridded monthly flux differences between the operational CERES-AQUA and CERES-NPP retrievals (similar as Figures 3, 5, 6) would provide an all-inclusive uncertainty that the current simulated uncertainty estimates could put into perspective.

Detailed comments:

1. The methodology description is very confusing. Please explain clearly how each of the three products are produced, use an acronym for each of them and the intermediate products, i.e.,

SNR- Aqua – simulated narrowband radiance with Aqua footprint size

SBR -Aqua – simulated broadband radiance with Aqua footprint size

SBF-Aqua --- simulated broadband flux with Aqua footprint size

2. Need a flow chart to illustrate the methodology. Something like:

MODIS-Aqua -> SNN-Aqua -> (narrowband to broadband conversion) SBR-Aqua

->(MODIS ADM) SBF-Aqua

MODIS-Aqua -> SNN-NPP -> SBR-NPP -> (MODIS ADM) SBF-NPP

MODIS-Aqua -> SNN-NPP -> SBR-NPP -> (like-VIIRS ADM) SBF-NPP2

3. Once product acronym is defined, specify which two products are compared in Figures 3, 5, 6.

4. Line 71-73 From this sentence, it is inferred that one of the CERES instrument on the TERRA/AQUA satellites used RAP scanning mode to create ADM at the beginning of the mission. How long was the RAP mode in operation? Line 82-83 says ADM was created from multi-year CERES measurements from both RAP and crosstrack mode. Does that mean both CERES instruments were used for ADM? Please clarify.

5. Line 119: why do you use different view zenith angle requirement for LW radiance?

6. Figure 1 and Table 1: could you add plots for total radiance comparison between CERES-NPP and CERES-AQUA? Also, the difference in radiances between CERES-NPP and CERES-NPP could be partly due to different footprint size besides the calibration?

7. Figure 2 Caption: add "CERES" before footprints

8. Line 170-173: The narrow-band to broad-band conversion coefficients are derived from simulated MODIS AQUA radiances and CERES-AQUA broadband radiance. Could these coefficients also be footprint size dependent? The same coefficients are now applied to convert both simulated CERES-AQUA and CERES-NPP footprint.

9. Line 177-178: Actually it would be interesting to compare the simulated CERES-AQUA and CERES-NPP fluxes at footprint level even with different footprint size. How would this comparison differ from the comparison between matched CERES-AQUA and CERES-NPP observations in Figure 1?

10. Figure 4 (cloud property difference between MODIS and VIIRS) and the generation of SBF-NPP2 are better introduced before comparison the three products (Figure 3. 5, 6) to ease the logic flow. Also be specific about difference (MODIS- VIIRS) or the other way?

11. Paragraph: 238-250 how is uncertainty defined?

12. Could you provide a map of difference in ADM?

13. Finally, please have native English-speaking colleagues check the grammar usage. There are many awkward sentences.

---

## Referee Comment (RC3) · Anonymous Referee #2 · 14 Jun 2017

Review

**The effects of di_erent footprint sizes and cloud algorithms on the top-of-atmosphere radiative flux calculation from CERES instrument on Suomi-NPP**
**By Wenying Su et al,**

This paper studies the effects of different footprint sizes and cloud algorithms on the radiative flux calculation from CERES on Suomi-NPP. Quantifying uncertainties from different footprint sizes and cloud algorithms in CERES-NPP is an important issue in calibration.

However, major revision of the text is needed. The sources of uncertainties in these two CERES (CERES-NPP and CERES-Aqua) observations are not fully explained in the text. The collocated radiances themselves already have 1.5%, 0.5%, and 0.1% of intrinsic differences for each channel, respectively. The sampling errors from two different footprint sizes are also embedded in the radiance comparison.

This study assumes that CERES-NPP and CERES-Aqua are identical instruments with compatible performance. But one may ask if there were improvements in CERES-NPP instrumentation/electronics or calibration. Possible degradation of CERES-Aqua instrument can be mentioned, if any. Before discussing uncertainties from footprint sizes and cloud algorithms, comprehensive uncertainty analysis of two CERES instruments is necessary to quantify errors from spatial sampling and cloud algorithm differences. I understand that this work is the first step towards identification of the two different CERES observations, but further uncertainty analysis and detailed descriptions are required. The methods to identify the effects of different footprints and cloud algorithms need to be described in detail.

1. Introduction : Line 71, Could you provide detailed explanation how two instruments help to construct ADMs that readers can visualize what you described?
2. Introduction : Line 88, "These pixel-level cloud properties are spatially and temporally matched with the CERES footprints ~" ,
   What is the spatial and temporal window for scene type matching from MODIS?

3. Line 126-128,
   "These differences do not show any view zenith angle dependence".
   Figure 1 compares the radiances of SW, daytime LW, and nighttime LW between two CERES measurements, but does not mention view zenith angle difference. Please explain why these differences do not show any view zenith angle dependence. Daytime LW is derived as the difference between total and SW channels. Nighttime LW is the same quantity with the total radiance because SW radiance is zero during nighttime. How daytime LW difference can be lower than SW difference? Are they offset with bias?

4.  Table 1.
    The difference between two cases is largest in SW radiance, but it is less in day time LW radiance. Authors can explain the reason why the difference in daytime LW radiance, which is derived as the difference between total and SW channels.

    Line 212-14,
    "Polar region cloud fraction differences are mainly because (that) VIIRS lacks the water vapor and CO2 channels which affect the polar cloud mask algorithm." Please explain how lack of those channels cause polar region cloud fraction difference. While VIIRS shows less cloud fraction over northern high latitude snow regions, it shows more cloud fraction over arctic. If all positive and negative difference is caused by large uncertainty of VIIRS, then more explanations on VIIRS cloud retrieval limitations including usage of different parameterization over snow/ice surfaces and uncertainty caused by this difference should be mentioned.

    Line 158 are used to derived → are used to derive

    5. Conclusions

    299 How these simulations helped to calibrate CERES-NPP observations?
    306 How much is the uncertainties of CERES ADMS as appeared in Su et al (2015)?

    314-316 VIIRS and MODIS cloud retrieval algorithms are not consistent. Then it is important to quantify the differences and reflect them to improve the CERES-NPP algorithm. Last sentence, "To maintain the consistency of the CERES climate data record, it is thus important to maintain the consistency of cloud retrieval algorithms." does not help much to solve the problem.

---

## Author Comment (AC1) · 10 Jul 2017

Review #1

General comments:
1. Whereas the overall research is of great importance as the various EOS algorithms are moving into the JPSS era, this paper is plain and simple incomplete. The authors give a pretty good discussion of the footprint size difference estimation experiments, in the title and the abstract they speak of retrieval algorithm differences as well, yet in the text the treatment of retrieval algorithm barely gets a sentence or two. I feel like the entire discussion of the retrieval algorithm is missing. In research it is important for published result to be reproducible to at least some degree. This work is definitely not reproducible. The footprint difference evaluation needs more detail and the algorithm difference evaluation plain and simple needs to be written.
*Thank you for your time to review this paper. We added a paragraph on page 9 to provide more details on the cloud retrieval differences between MODIS and Aqua. We also revised the method section extensively with the new diagram (Figure 6) to illustrate the simulation process.*

2. Throughout, please consider the audience may not know what your different acronyms are. Please expand in text or provide an acronym table.
*We defined all the acronyms in the paper.*

Specific comments:

Line 7: CERES, first mention of the instrument, please expand
*We defined CERES in the title, which should give reader a reference.*

Line 12: same thing for VIIRS and MODIS, consider the audience may not know what the acronyms are.
*Defined.*

Line 26: "cloud property differences", elaborate what exact cloud properties you mean
*Defined the cloud property as "cloud fraction and optical depth".*

Line 48: "flied" is not a word
*Thank you for catching the typo, corrected.*

Line 49: "were launch" is grammatically incorrect
*Corrected in the revised version.*

Line 56-57: spacecrafts is not a word
*We replaced "spacecrafts" with "satellites".*

Lines 115-117: why have you not considered the scattering angle?
*The scattering angle is intrinsically considered here, as cos(scattering angle) = cos((SZA)cos(VZA) + sin(SZA)sin(VZA)cos(RAZ), where SZA is solar zenith angle, VZA is viewing zenith angle, and RAZ is relative azimuth angle.*

Line 128: "the nighttime LW radiance is simply from the total channel measurements" doesn't make any sense.

*We don't quite understand the reviewer's comment. As we stated in the paper, the CERES instrument measures radiances in shortwave, window, and total channels. For daytime measurements, the longwave radiance is derived as the difference between total channel and shortwave channel. For nighttime measurements, the shortwave radiance is zero thus the longwave radiance is the same as the total channel. We changed this sentence to "the nighttime LW radiance is directly derived from the total channel measurements" to be consistent with the previous sentence.*

Line 157: snow /no snow conditions, how are you getting this information?

*Snow information was obtained from the national snow ice data center and is included in the CERES SSF data. Detailed description is added on page 5.*

Line 159: "to derived" is grammatically incorrect

*Corrected, thank you!*

Line 163-166: how are you getting snow/no snow information?

*Snow information was obtained from the national snow ice data center and is included in the CERES SSF data. Detailed description is added on page 5.*

Line 178: replace 'assessed' with 'ascertained'

*Changed.*

You talk about the simulated footprint generation, but where is the discussion of the algorithm differences? "We tweaked it" is not good enough.

*The cloud algorithm difference between MODIS and VIIRS cloud retrieval are discussed on page 9. However, please note that addressing the cloud retrieval difference is not the focus of this paper. The main focus of the paper is to address the effect of footprint size and cloud retrieval differences on flux inversion. We don't understand where "We tweaked it" is coming from, as we didn't use any wordings like that in the paper and therefore unable to address the reviewer's comment.*

Line 192: "that consist with" is not English, "result in less footprints" is not either.

*We modified the sentence to "as the narrowband-to-broadband regressions are only applied to footprints that consist of the same surface types which result in fewer footprints with valid fluxes for CERES-NPP than for CERES-Aqua."*

Line 204-210: Why? Please explain

*Explanations of the cloud fraction and cloud optical depth differences were given on lines 197-205.*

Line 224-227: This does not make any sense to anyone who might theoretically even consider checking your results.

*Readers who might want to check our results can download the CERES-Aqua and CERES-NPP SSF data and calculate the daily ratio for each 1 degree by 1 degree grid box using the cloud fraction and cloud optical depth of the CERES footprints.*

Line 235: Why did your flux increase? Explain

*CERES flux is derived as: π\*radiance/anisotropic factor (see Equation 1 in the revised manuscript). As we explained in the paper, for oblique viewing angles, the anisotropy factors for thick clouds are smaller than those for thin clouds. As anisotropy factors decrease when retrieval cloud optical depths increase, the inverted fluxes increase (as the radiances didn't change). We understand this part might be confusing, to help reader better understand how we derive flux from CERES radiance measurements, we added some descriptions in the introduction section.*

Line 273: replace 'is' with 'are'
*Modified.*

Line 275: replace 'these' with 'that'
*We rewrote this sentence.*

Line 282: replace 'thus' with 'to'
*Changed.*

Comments on figures:

Figure 2: so these CERES footprints are misaligned like that in real life or just in illustration. Please clarify.
*We rewrote the paper, and Fig. 2 is replaced with the flow chart (Fig. 6). The original Fig. 2 is just for illustration, not how CERES footprints look like in real life.*

Figures 3-6: can't be evaluated because I don't know what I'm looking at. There is no explanation in the text as to why the images look they way they do. Could be a thousand different reasons, so figures do not make sense.
*We modified the paper and the captions of these figures to make it clear to readers.*

---

## Author Comment (AC2) · 10 Jul 2017

Review #2

Major comments

This manuscript attempts to quantify the flux uncertainties in the NPP-CERES fluxes because NPP does not have a rotating CERES instrument to generate NPP specific ADM as the TERRA and AQUA satellites had and has to borrow the ADM generated from AQUA satellite. The study uses AQUA MODIS radiances to create simulated fluxes for both AQUA and NPP footprints - going through the same narrowband to broadband conversion and using the same ADM for radiance to flux conversion to isolate the differences due to footprint size alone. The differences due to both footprint size and cloud properties are examined with VIIRS-adjusted cloud properties and subsequent ADM. This study helps to understand and quantify the uncertainty in the current NPP CERES fluxes due to missing its own ADM. However, the description of the methodology could be much improved by adding proper flow chart and naming convention.
*Thank you for your time reviewing this paper. We added a flow diagram (Figure 6) to illustrate the process of convoluting the high-spatial-resolution MODIS pixel-level radiances and cloud/aerosol retrievals with CERES-Aqua and CERES-NPP point spread functions to provide narrowband radiances and cloud/aerosol properties for the respective simulated CERES-Aqua and CERES-NPP footprints, and the simulated CERES-NPP footprints with VIIRS-like cloud properties. These narrowband radiances are then used to derive broadband radiances and they are converted to fluxes using the Aqua ADMs.*

The other missing piece of information is how the uncertainties estimated with simulated fluxes starting from MODIS pixel radiances are measured against actual retrieved NPP CERES flux?
*As illustrated in Fig. 3 in the revised manuscript, the overpass times for NPP are different from those of Aqua. The broadband radiances generated using narrowband-to-broadband coefficients also differ from the measured broadband radiances. Direct comparison between fluxes in the simulated CERES-NPP footprints and the actual CERES-NPP footprints will mostly show the difference caused by the afore mentioned differences (the simulated CERES-NPP used the Aqua Sun-viewing geometry). The focus of the paper is to assess the effect of footprint size difference and scene identification difference on flux inversion without having to account for the overpass time difference and calibration difference. This is the reason that we designed this study and used narrowband-to-broadband regression to derive the broadband radiances.*

Similar comparisons between the matched NPP and AQUA radiances (Figure 1) could be applied to match fluxes. Likewise, gridded monthly flux differences between the operational CERES-AQUA and CERES-NPP retrievals (similar as Figures 3, 5, 6) would provide an all-inclusive uncertainty that the current simulated uncertainty estimates could put into perspective.
*We compared the fluxes using matched NPP and Aqua footprints as suggested by the reviewer. Results are provided in Fig. 2 and Table 1, and descriptions are on page 8. However, as mentioned above and demonstrated in Fig. 3, the overpass time differences between the Aqua and NPP orbits dominate the monthly gridded flux differences, as shown below. The left plot*

*shows that monthly gridded SW flux for Aqua and the right plot shows the SW flux differences between NPP and Aqua. Most of the features in the difference plot resemble those in Fig. 3.*

[Figure]

*We can compare the albedo between NPP and Aqua to bypass the insolation difference, as shown below (figure on the left is the monthly mean albedo and figure on the right is albedo difference between NPP and Aqua). The global monthly mean albedo from CERES-NPP is about 0.003 (1.02%) greater than the global monthly mean albedo from CERES-Aqua (0.292). We believe these differences stem from the combination of calibration difference, ADM and scene type differences. It is impossible to attribute how much each of them contributed to the albedo difference just by comparing the flux/albedo from Aqua and NPP. This is the reason that we come up with this study to isolate the contribution of ADM and scene type differences to flux difference.*

*We added the albedo difference plot (Fig. 4) in the revised manuscript and some discussions on page 8.*

[Figure]

Detailed comments:

1. The methodology description is very confusing. Please explain clearly how each of the three products are produced, use an acronym for each of them and the intermediate products, i.e.,

SNR- Aqua - simulated narrowband radiance with Aqua footprint size
SBR -Aqua - simulated broadband radiance with Aqua footprint size
SBF-Aqua --- simulated broadband flux with Aqua footprint size

*We rewrote the methodology section and defined that $I_a^s(\lambda)$ is the narrowband radiance for the simulated CERES-Aqua footprints, $I_a^s$ is the broadband radiance for the simulated CERES-Aqua footprints, and $F_a^s$ is the broadband flux for the simulated CERES-Aqua footprints. Their counterparts for the simulated CERES-NPP footprints are $I_n^s(\lambda)$, $I_n^s$, and $F_a^s$, and they are $I'^s_n(\lambda)$, $I'^s_n$, and $F'^s_a$ for the simulated CERES-NPP footprints with VIIRS-like cloud fraction and*

cloud optical depth. The CERES-Aqua measured broadband radiance is $I_a^m$, and the CERES-NPP measured radiance is $I_n^m$. The corresponding fluxes are $F_a^m$ and $F_n^m$.

2. Need a flow chart to illustrate the methodology. Something like:
MODIS-Aqua -> SNN-Aqua -> (narrowband to broadband conversion) SBR-Aqua ->(MODIS ADM) SBF-Aqua
MODIS-Aqua -> SNN-NPP -> SBR-NPP -> (MODIS ADM) SBF-NPP
MODIS-Aqua -> SNN-NPP -> SBR-NPP -> (likeNIiRS ADM) SBF-NPP2
*We added a flow diagram (Figure 6) to illustrate the process of convoluting the high-spatial-resolution MODIS pixel-level radiances and cloud/aerosol retrievals with CERES-Aqua and CERSE-NPP point spread functions to provide narrowband radiances and cloud/aerosol properties for the respective simulated CERES-Aqua, and simulated CERES-NPP footprints with MODIS cloud properties and with VIIRS-like cloud properties. These narrowband radiances are then used to derive broadband radiances and they are converted to fluxes using Aqua ADMs.*

3. Once product acronym is defined, specify which two products are compared in Figures 3, 5, 6.
*Modified the captions using the acronym (now Figures 7, 8, 9).*

4. line 71-73 From this sentence, it is inferred that one of the CERES instrument on the TERRA/AQUA satellites used RAP scanning mode to create ADM at the beginning of the mission. How long was the RAP mode in operation? line 82-83 says ADM was created from multi-year CERES measurements from both RAP and crosstrack mode. Does that mean both CERES instruments were used for ADM? Please clarify.
*We specified the total lengths of RAP data collect for Terra and Aqua on lines 83-84: "There are about 60 months of RAP data collected on Terra and about 32 months of RAP data collected on Aqua.". We clarified that the "Terra ADMs and Aqua ADMs were developed separately using multi-year CERES Terra and Aqua measurements in RAP mode and in cross-track mode using the scene identification information from MODIS" at lines 98-101.*

5. line 119: why do you use different view zenith angle requirement for LW radiance?
*We modified the SW radiance matching criterion of viewing zenith angle to be the same as LW radiance (i.e. 2 degrees). We updated the Table 1 and Figure 1 for the SW comparison.*

6. Figure 1 and Table 1: could you add plots for total radiance comparison between CERES-NPP and CERES-AQUA? Also, the difference in radiances between CERES-NPP and CERES-NPP could be partly due to different footprint size besides the calibration?
*CERES instruments measure radiances in total, SW, and window channels. These measured radiances are filtered radiances, which then are unfiltered by accounting for the spectral response function for each channel. Unfiltered radiances are provided for LW, SW, and window channels, but not for total channel, thus we are unable to compare the total radiances. The reviewer is correct, that footprint size difference also contributes to the radiance difference. However, the effect of footprint size on radiance should be random. The radiance bias is mostly from the calibration difference between CERES-Aqua and CERES-NPP, and the footprint size difference can increase the RMS error. We added this in the revised version.*

7. Figure 2 Caption: add "CERES" before footprints
*This figure is replaced with Figure 6 in the revised version.*

8. line 170-173: The narrow-band to broad-band conversion coefficients are derived
from simulated MODIS AQUA radiances and CERES-AQUA broadband radiance. Could
these coefficients also be footprint size dependent? The same coefficients are now
applied to convert both simulated CERES-AQUA and CERES-NPP footprint.
*The narrowband-to-broadband (NB2BB) regression coefficients should not be dependent on the*
*footprint size, but the broadband and narrowband radiances and fluxes could change as the*
*footprint size changes which sometimes results in more/fewer clouds in the footprints.*

9. line 177-178: Actually it would be interesting to compare the simulated CERES-AQUA
and CERES-NPP fluxes at footprint level even with different footprint size. How would
this comparison differ from the comparison between matched CERES-AQUA and CERESNPP
observations in Figure 1?
*We compared fluxes using matched CERES-Aqua and CERES-NPP footprints in Figure 2 and*
*also included the mean fluxes and root-mean-square errors in Table 1. Some discussions are*
*also added to the paper (page 8). We also compared the fluxes from simulate CERES-Aqua and*
*simulated CERES-NPP at the footprint level using data of April 1, 2013. Flux differences and*
*RMS errors are provided on page 12, which are much smaller than the flux biases and RMS*
*errors listed in Table 1.*

10. Figure 4 (cloud property difference between MODIS and VIIRS) and the generation of
SBF-NPP2 are better introduced before comparison the three products (Figure 3.5, 6) to
ease the logic flow. Also be specific about difference (MODIS- VIIRS) or the other way?
*We revised the paper by adding a section dedicated to the comparison between NPP and Aqua*
*(section 2) and the cloud property differences are discussed in this section in the revised*
*manuscript (Figure 5).*

11. Paragraph: 238-250 how is uncertainty defined?
*We revised this paragraph to be more specific. The uncertainty refers to the differences between*
*simulate CERES-Aqua and CERES-NPP.*

12. Could you provide a map of difference in ADM?
*As shown in Equation (1), for a given scene type, ADM (R) is defined as a function of solar*
*zenith angle, viewing zenith angle, and relative azimuth angle. Additionally, ADM is also*
*different for different scene types. Averaging anisotropy values into monthly grid box means*
*could be misleading.*

13. Finally, please have native English-speaking colleagues check the grammar usage.
There are many awkward sentences.
*We did a thorough editing of the paper.*

---

## Author Comment (AC3) · 10 Jul 2017

Review #3

This paper studies the effects of different footprint sizes and cloud algorithms on the radiative flux calculation from CERES on Suomi-NPP. Quantifying uncertainties from different footprint sizes and cloud algorithms in CERES-NPP is an important issue in calibration.

However, major revision of the text is needed. The sources of uncertainties in these two CERES (CERES-NPP and CERES-Aqua) observations are not fully explained in the text. The collocated radiances themselves already have 1.5%, 0.5%, and 0.1% of intrinsic differences for each channel, respectively. The sampling errors from two different footprint sizes are also embedded in the radiance comparison.

*The actual CERES-Aqua and CERES-NPP measured radiances differ, as mentioned above. These comparisons are based upon matched CERES footprints from Aqua and NPP, thus the sample number used here are the same for Aqua and NPP. The footprint size differences can contribute to the radiance differences, but these differences should be random. We believe that the footprint size differences can increase the RMS errors, but the mean radiance differences are mostly resulted from calibration differences between CERES-Aqua and CERES-NPP (page 7). The goal of the simulation designed in this paper is to isolate the effect of footprint size difference on flux inversion without having to unravel the contribution of calibration difference to the flux difference.*

This study assumes that CERES-NPP and CERES-Aqua are identical instruments with compatible performance. But one may ask if there were improvements in CERES-NPP instrumentation/electronics or calibration. Possible degradation of CERES-Aqua instrument can be mentioned, if any. Before discussing uncertainties from footprint sizes and cloud algorithms, comprehensive uncertainty analysis of two CERES instruments is necessary to quantify errors from spatial sampling and cloud algorithm differences. I understand that this work is the first step towards identification of the two different CERES observations, but further uncertainty analysis and detailed descriptions are required. The methods to identify the effects of different footprints and cloud algorithms need to be described in detail.

*Instrument degradation is indeed an issue if it is not accounted in the calibration. CERES instruments are designed with onboard calibration and instrument degradation is accounted for when the monthly gain factors are derived. We provided some discussion of the instrument gains on page 7. The calibration differences between CERES-Aqua and CERES-NPP are not fully understood yet. The team is working on to understand the differences and eventually place both of them on the same radiometric scale. The focus of the paper is not to address the calibration difference. The simulated CERES-NPP footprints are designed to quantify the flux uncertainty due to footprint size difference and cloud fraction and cloud optical depth differences without having to unravel them from the calibration difference.*

1. Introduction: Line 71, Could you provide detailed explanation how two instruments help to construct ADMs that readers can visualize what you described?

*One of the CERES instrument is placed in RAP mode, which is designed to maximize the angular coverage for constructing ADMs, while the other one is placed in cross-track mode to provide*

*daily global coverage. The objective here is to use one instrument to maximize the angular coverage without having to sacrifice the spatial coverage. We rewrote this part in the revised manuscript to make this clear.*

2. Introduction: Line 88, "These pixel-level cloud properties are spatially and temporally matched with the CERES footprints ~" ,
What is the spatial and temporal window for scene type matching from MODIS?
*The spatial matching is to include all MODIS pixels (1 km) in the CERES footprint (20 km at nadir). The temporal matching is within 20 seconds when CERES is in cross-track mode, and is within 6 minutes when CERES is in RAP mode. We specified these criteria used on page 5.*

3. Line 126-128,
"These differences do not show any view zenith angle dependence".
Figure 1 compares the radiances of SW, daytime LW, and nighttime LW between two CERES measurements, but does not mention view zenith angle difference. Please explain why these differences do not show any view zenith angle dependence. Daytime LW is derived as the difference between total and SW channels. Nighttime LW is the same quantity with the total radiance because SW radiance is zero during nighttime. How daytime LW difference can be lower than SW difference? Are they offset with bias?
*The comparisons shown in Figure 1 include data taken from nadir viewing angles to oblique angles with viewing zenith angle greater than 60 degrees. When we compare the radiances taken at different viewing zenith angle ranges (e.g. <20 degrees, >20 degrees, >40 degrees), the RMS errors of the radiances remain almost the same. This indicates that these differences do not show any viewing zenith angle dependence. We added these discussions in the revised manuscript (page 7).*

*The reason that the daytime LW difference is smaller than that of SW is because the difference of the total channel is slightly smaller than the difference of the SW channel. When the SW radiance is subtracted from the total radiance, the difference in LW is not as large as that of SW.*

4. Table 1.
The difference between two cases is largest in SW radiance, but it is less in day time LW radiance. Authors can explain the reason why the difference in daytime LW radiance, which is derived as the difference between total and SW channels.
*As mentioned above, the reason that the daytime LW difference is smaller than that of SW is because the difference of the total channel is slightly smaller than the difference of the SW channel. When the SW radiance is subtracted from the total radiance, the difference in LW is not as large as that of SW. We added some explanation on Page 7.*

Line 212-14,
"Polar region cloud fraction differences are mainly because (that) VIIRS lacks the water vapor and CO2 channels which affect the polar cloud mask algorithm."
Please explain how lack of those channels cause polar region cloud fraction difference. While VIIRS shows less cloud fraction over northern high latitude

snow regions, it shows more cloud fraction over arctic. If all positive and negative difference is caused by large uncertainty of VIIRS, then more explanations on VIIRS cloud retrieval limitations including usage of different parameterization over snow/ice surfaces and uncertainty caused by this difference should be mentioned.

*The CERES cloud working group developed sophisticated cloud detection algorithms using visible and infrared channels of MODIS separately for polar and non-polar regions and for daytime, twilight, and nighttime. However, these detection algorithms have to be modified to apply to the VIIRS observations, as some of the MODIS channels utilized for cloud detection are not available on VIIRS. These modifications include replacing the 2.1 um MODIS channel with the 1.6 um VIIRS channel, and replacing detection tests using MODIS 6.7 um and 13.3 um channels with VIIRS 3.7 um and 11 um channels, and supplement with tests utilizing VIIRS 1.6 um channel and the brightness temperature differences between 11um and 12um. These changes mainly affect the cloud detections over the polar regions, and as different tests and thresholds are used for daytime and twilight cloud detections, these changes will also affect different latitude zones differently.*

Line 158 are used to derived → are used to derive
*Corrected, thank you!*

5. Conclusions
299 How these simulations helped to calibrate CERES-NPP observations?
*These simulations are not designed to calibrate the CERES-NPP radiance observations. The focus of the paper is to assess the effect of footprint size difference and scene identification difference on flux inversion without having to account for the overpass time difference and calibration difference.*

306 How much is the uncertainties of CERES ADMS as appeared in Su et al (2015)?
*The biases and RMS errors provided in Su et al. (2015) is for diurnally averaged SW and LW flux. The bias for regional monthly mean TOA SW flux is less than 0.2 Wm-2 and the RMSE is less than 1.1 Wm-2. For TOA LW flux, the regional monthly mean bias is less than 0.5 Wm-2 and the RMSE is less than 0.8 Wm-2.*

314-316 VIIRS and MODIS cloud retrieval algorithms are not consistent. Then it is important to quantify the differences and reflect them to improve the CERESNPP algorithm. Last sentence, "To maintain the consistency of the CERES climate data record, it is thus important to maintain the consistency of cloud retrieval algorithms." does not help much to solve the problem.
*Figure 5 quantify the cloud fraction and cloud optical depth difference between Aqua and NPP. The CERES team developed cloud retrieval algorithm using MODIS observations for over ten years. Because VIIRS lacks some of the channels that were used for MODIS cloud retrieval, to make the cloud retrieval consistent between VIIRS and MODIS, the MODIS and VIIRS retrieval algorithms have to be reevaluated using only channels that are available for both MODIS and VIIRS. This is a grand challenge and will take time to develop the new algorithms and to process*

*data of more than 30 instrument years. We rewrote this sentence to reflect the need to develop algorithm considering the capabilities of both MODIS and VIIRS.*

---

## Referee Report (RR1)

**Overall comment:**

The manuscript has much improved from the first version I reviewed. However I still have a few concerns that I would like the authors to address.

**Overall recommendation:**

Publish pending revision.

Major comments:

Beginning of Section 2: you guys still haven't explained why you didn't use the scattering angle differences in your pixel matching algorithm. It's the scattering angle that really matters when comparing VIIRS and MODIS. When you do individual angle thresholds the differences can add up constructively and you end up with a large scattering angle difference, larger than you'd want to have in any case.

Figure 3: Can you please explain the source of the dark blue band of lower insolation around 60 deg. N? That almost looks like some kind of an artifact. Based on that banding artifact, your discussion on Page 8 that TOA fluxes track insolation differences is not valid. The TOA fluxes do not display any banding at 60 deg. N.

Minor comments:

Line 14: "… the imagers sharing the spacecraft with CERES-NPP… ."

Line 91: reference for the IGBP dataset?

Line 98: Do you mean "VIIRS"? If not, please expand what VIRS stands for.

Line 200: "… to be applicable to the VIIRS observations…."

Line 207: "…. was generated …. "

Line 245: " … except for small differences … "

Line 365: "… the imagers sharing the spacecraft … " They are mounted together, not flying in a constellation

Line 386: " … five years …." no dash

---

## Author Response (AR2)

Reviewer 1:

The revised manuscript has made significant improvements. A few remaining questions for clarification:

Line 86-90: Are cloud properties retrieved on its native MODIS/VIIRS resolution and then converted into CERES footprint or the radiances are converted into CERES footprint resolution first and then applied with cloud retrieval algorithm? If the latter, mention radiance then cloud property.

*The cloud properties are retrieved on the native MODIS/VIRRS resolution, then averaged into the CERES footprint.*

Line 164-167: The FM3 and FM5 response to LW seems to explain the NPP LW radiance smaller compared to Aqua LW radiance. But SW responses of FM3 and FM5 seem contradictory to the observed radiance differences.

*What we discussed on lines 164-167 is the normalized instrument gains for each CERES FM3 and FM5 since the beginning of the mission. The changes provided are relative to the beginning of the mission for each instrument, or the relative changes over the years. The relative change of each instrument is different from the absolute differences shown in Figure 1.*

Figure 6: Based on my understanding, VIIRS-like cloud would only affect the choice of ADM, not the generation of NB radiance for NPP footprint size and the conversion from NB to BB radiance. If this is the case, the right branch of the procedure for NPP with VIIRS-like retrieval is not accurate. Do step one and step two affected by cloud?

*The reviewer is correct, the VIIRS-like cloud would only affect the selection of ADMs, but not the MODIS NB radiances and the conversion from NB to BB radiances. Figure 6 is modified accordingly and text describing the figure is also modified. Thank you for pointing this out.*

Reviewer 2:

Overall comment:

The manuscript has much improved from the first version I reviewed. However I still have a few concerns that I would like the authors to address.

Major comments:

Beginning of Section 2: you guys still haven't explained why you didn't use the scattering angle differences in your pixel matching algorithm. It's the scattering angle that really matters when comparing VIIRS and MODIS. When you do individual angle thresholds the differences can add up constructively and you end up with a large scattering angle difference, larger than you'd want to have in any case.

Please evaluate the scattering angle differences between MODIS and VIIRS in your selected footprints. Differences in individual zenith/azimuth angles can add up constructively and create significant scattering angle differences.

*As we explained in our first round response, cos(scattering angle) = cos((SZA)cos(VZA) + sin(SZA)sin(VZA)cos(RAZ). The SW radiance matching criteria used for Fig. 1 and Table 1 are that latitude and longitude differences are less than 0.05 degree, and the SZA difference is less than 2 degrees, VZA difference is less than 2 degrees, and the relative azimuth angle difference is less than 5 degrees. To answer the reviewer's concern, we calculated the scattering angle differences between matched Aqua and NPP footprints. The PDF of the scattering angle differences is shown below on the left and the scattering angle differences are less than 2° for about 95.6% of the matched footprints and are less than 3° for about 99.9% of the matched footprints. We added on page 7 that the criteria we used also provide tight constraint on scattering angle. When we added the scattering angle difference<2° to the matching criteria, the SW result (shown below on the right) is very similar to that in Figure 1 and Table 1 and don't change any conclusions in the paper (added on page 7).*

[Figure]

*To further illustrate that the SZA/VZA/RAZ matching criteria offer better angular matching than using scattering angle, we modified the SW footprint matching algorithm by using latitude and longitude differences less than 0.05 degree and scattering angle difference less than 2 degrees.*

*The comparison of matched Aqua and NPP footprints are shown below. One thing we notice is that the number of matched footprints increased from 147,894 (Table 1) to 386,471, and the RMSE also increased from 4.1 Wm-2sr-1 to 4.7 Wm-2sr-1.*

[Figure]

*The reason that the number of matched footprints increased so much is that using scattering angle difference <2° to match footprints can introduce some large VZA and RAZ differences. Plots below show the PDFs of VZA and RAZ differences when the scattering angle differences are less than 2 degrees (the SZA differences are always less than 0.5 degrees as the data we used are when both satellites flew in tandem). There are significant number of matched footprints have VZA differences greater than 2 degrees and RAZ differences greater than 5 degrees. The table below shows three case where scattering angle differences are less than 2 degrees, but the RAZ difference can be 40 degrees (first case), the VZA difference can be 20 degrees (last case).*

[Figure]

| SZA | 56.5 | 56.4 | 50.0 | 50.0 | 23.5 | 23.5 |
|------|-------|-------|------|------|------|------|
| VZA | 0.1 | 2.3 | 43.1 | 32.0 | 28.2 | 6.3 |
| RAZ | 157.1 | 116.3 | 71.5 | 74.0 | 51.2 | 52.5 |
| Scat | 56.6 | 57.4 | 50.5 | 48.9 | 22.1 | 20.3 |

*We hope the above explanation can help the reviewer understands why we use SZA/VZA/RAZ angles to match the footprints instead of scattering angles.*

Figure 3 appears to contain an artifact such that it makes statements made in the paper invalid. Either correct the figure or explain the discrepancy / source of banding.
Figure 3: Can you please explain the source of the dark blue band of lower insolation around 60 deg. N? That almost looks like some kind of an artifact. Based on that banding artifact, your discussion on Page 8 that TOA fluxes track insolation differences is not valid. The TOA fluxes do not display any banding at 60 deg. N.

*Figure 3 shows the monthly mean TOA insolation differences (NPP-Aqua), the monthly mean TOA reflected SW flux difference plot was not included in the paper, but was included in our first round response (shown below for reference). The TOA reflected SW flux difference plot clearly shows the same "dark blue" band as that in Figure 3. We don't understand the reviewer's statement "The TOA fluxes do not display any banding at 60 deg. N.", as the "dark blue" band is definitely there. Maybe the reviewer is mistaken the albedo difference (Figure 4) for TOA flux difference.*

[Figure]

*The "dark blue" bands in both Figure 3 and the figure above are not an artifact, they are caused by the orbit differences between Aqua and S-NPP. Plots below shows the SZA density distributions for CERES-Aqua and CERES-NPP for different latitude zones for April 2013. At 60° N, the NPP overpasses at a lot more SZA>70 degrees than Aqua, which causes the solar insolation for NPP overpass time being smaller than that for Aqua, hence the TOA reflected SW flux (and this is why the "dark blue" band was not visible for the albedo plot). We modified a sentence on page 8 to reflect this fact: "Solar insolation for NPP overpass times are greater than that for Aqua overpass times over most regions, except over the northern high latitude where NPP has significantly more overpasses at $\theta_0>70°$ than Aqua".*

[Figure]

Minor comments:

Line 14: "… the imagers sharing the spacecraft with CERES-NPP… ."
*Modified.*

Line 91: reference for the IGBP dataset?
*Added.*

Line 98: Do you mean "VIIRS"? If not, please expand what VIRS stands for.
*Here VIRS (the Visible and Infrared Scanner) is the correct imager, it is already defined on line 51.*

Line 200: "… to be applicable to the VIIRS observations…."
*Modified.*

Line 207: "…. was generated …. "
*Modified.*

Line 245: " … except for small differences … "
*Modified.*

Line 365: "… the imagers sharing the spacecraft … " They are mounted together, not flying in a constellation
*Modified as "there are some differences between the imagers that are on the same spacecrafts as CERES-Aqua (MODIS) and CERES-NPP (VIIRS)".*

Line 386: " … five years …." no dash
*Modified to "five years".*

[revised manuscript text omitted]

---

## Author Response (AR3)

Thank you for your revision. After some follow-up discussion with a previous Reviewer, they are concerned that you did not explain the high-latitude artefact in Figure 3 sufficiently. Following on from their initial review comments, could you please check whether this is a plotting error, and if it is not, go in to more detail about this artefact? This is a major sticking point of theirs before they can assess the revision in detail, as they feel that if the artefact cannot be explained then the later analysis may not hold up.

Figure 3 is correct and the "dark blue" band is not an artifact, it is caused by the difference in Aqua and NPP orbit. The figure below shows the monthly mean SZA differences between NPP and Aqua. For the 'dark blue' band in the solar insolation difference plot, the monthly mean SZA from NPP is about 4 degrees greater than that from Aqua orbit (which results in smaller insolation compared with Aqua). Over the tropical regions, the monthly mean SZA from NPP is smaller than that from Aqua by as much as 3 degrees. This figure should help explain the differences in solar insolation better.

[Figure]

To help reviewers and readers understand the differences in the solar insolation, we added the TOA reflected SW flux differences between CERES-NPP and CERES-Aqua to the manuscript and modified the sentence on lines 193-195 to the following:

"When we compare the monthly gridded TOA reflected SW flux between CERES-NPP and CERES-Aqua (Figure 4), the difference features in high latitude regions (north of 60N and south of 60S) resemble those of the insolation differences."

---

## Author Response (AR4)

The major concern is as follows: line 200 states that a 0.003 (or 1.02%) difference is caused by calibration, with a reference to Figure 1, although this does not seem to be directly supported by the contents of Figure 1.

The suggestion is to add some statistics to the panels of Figure 1 (and Figure 2 as well). For example, mean offset, RMSE, and fit coefficients (i.e. y=ax+b). Presumably b will come out to 0 and a will come out to 1.0102 or close, or the mean bias will be around 0.003, which will then support your argument. (Or possibly the other direction dependent on whether Aqua or NPP is taken as the reference.) If not, then further discussion or amending the statement is required.

*We did provide the radiance and flux statistics in our paper in Table 1, which summarized the mean radiances and fluxes from CERES-NPP and CERES-Aqua, and the radiance and flux RMS errors as well. To make the sentence on Line 200 more clear we referenced to both Figures 1a and Table 1 in the revised version. We also added that the radiance and flux statistics are provided in Table 1 in the captions of Figure 1 and Figure 2.*

Minor comments:
line 1: please expand NPP, first use of the acronym
*Expanded NPP as National Polar-orbiting Partnership.*

line 25: "...when compared..."
*Modified.*

line 41: "...data product crucial to..."
*Modified.*

line 147: " … 2 degrees and 3 degrees respectively. ..."
*Modified.*

line 188: " … for April 2013. Insolation for NPP..."
*Modified.*

line 189: "… is greater than..."
*Modified.*

line 192: strike "solar". The word insolation means "solar radiation". You shouldn't double up.
*Change all solar insolation to insolation.*

Line 196: "… We then compare…"
*Changed.*

line 212: " … mainly affect cloud detections…"
*Modified.*

line 187, 213, 532, and everywhere else: remove "solar" when referring to "insolation"
*Changed.*

Fig 3. caption: should read "… mean insolation…"
*Modified.*